# Addressing Thermal Comfort in Regional Energy Poverty Assessment with Nussbaumer's MEPI

**Tiare Robles-Bonilla** [1,*] and **Karla G. Cedano** [2]

1   Centro de Investigación en Ingeniería y Ciencias Aplicadas, Universidad Autónoma del Estado de Morelos, Ave. Chamilpa 1001, Cuernavaca CP 62209, Mexico
2   Instituto de Energías Renovables-UNAM, Xochicalco, Azteca, Temixco CP 62588, Mexico; kcedano@ier.unam.mx
*   Correspondence: tiare.roblesbno@uaem.edu.mx; Tel.: +52-1735-206-9012

**Abstract:** Research on energy poverty (EP) started in the United Kingdom and other Western European countries in response to the Oil Crisis in 1973. In the last few years, the European community has made important breakthroughs on the topic, by establishing clear terminology as well as funding different multidisciplinary and intersectoral task groups that have EP understanding and alleviation as their goal. Several different methodologies have been developed to measure EP. For instance, the multidimensional energy poverty index (MEPI) by Nussbaumer et al. (2012) has been successfully used in Africa and in seven Latin American countries. Mexico does not have an official measure, indicator, or index on EP. However, a very important energy service has been overlooked: thermal comfort. In the present work, MEPI was understood as an energy services deprivation calculation, and thermal comfort was included. Understanding the regional nature of thermal comfort, we searched for weather-based regionalizations that could address a whole country diversity. We applied two regionalizations, one strongly related to political divisions (called climatic), and a another used for household design and construction standards (bioclimatic). The bioclimatic regionalization had a better fit when assessing energy services deprivation, since it addresses exclusively geographical and weather conditions, instead of the artificial political divisions. Having better ways to assess the level of EP in the local context is a key factor to develop effective public policies that might alleviate EP in a sustainable way.

**Keywords:** energy deprivation; thermal comfort; regionalization; bioclimates





## 1. Introduction

The topic of energy poverty, previously known as fuel poverty, emerged in the 1970s [1], with the purpose of addressing the lack of thermal comfort that was present in some European regions. Although the UK has been the country with the most studies, interest is growing in more regions of Europe, Africa, and Latin America. Energy poverty can be defined "as occurring when a household is incapable of securing a degree of domestic energy services (such as space heating, cooling, cooking) that would allow them to fully participate in the customs and activities that define membership in society" [2]. Literature shows that there are three main ways to evaluate energy poverty. First is the expenditure approach: "where examinations of the energy costs faced by households against absolute or relative thresholds provide a proxy for estimating the extent of domestic energy deprivation" [3]. In this sense, definitions coined in the 1990s referring to income make sense, such as the definition proposed by Boardman (1991) where she highlights that energy poverty is due to low income and the use of inefficient equipment [4]. However, a government funded review showed that the 10% measure was too sensitive to energy prices and so fluctuated a lot irrespective of the actual progress made to address important drivers such as energy efficiency of equipment and properties [3,5,6], rendering that approach obsolete. Also, this approach leads to confusion when classifying households as poor due to low

energy consumption, without taking into consideration that "poverty" could reflect an energy-saving household behavior. The second way to measure EP is the consensual approach: "based on self-reported assessments of indoor housing conditions, and the ability to attain certain basic necessities relative to the society in which a household resides" [3]. Most studies that seek to assess energy poverty use this approach, including this work. Lastly, there is the direct measurement approach: "where the level of energy services (such as heating) achieved in the home is compared to a set standard" [3]. This approach is not very frequently used since there are several technical and ethical problems in measuring and monitoring energy services and direct household income [3].

Nussbaumer et al. (2012) developed a multidimensional energy poverty index (MEPI) for Africa, centered around the lack of modern energy services. It captures both the incidence and intensity of energy poverty, thus providing a new tool to aid the development of public policies [7]. In this index, Nussbaumer et al. classified the countries according to their level of energy poverty, from acute energy poverty to moderate energy poverty. One of the things that Nussbaumer et al. observed was the lack of attention in the quality, reliability, and affordability of the energy services of a household. This is related to Modi et al., (2005) where energy services are defined as "the benefits that energy carriers produce for human wellbeing" [8].

In Mexico, the 36th article of Ley General de Desarrollo Social (LGDS) establishes that the Consejo Nacional de Evaluación de la Política de Desarrollo Social (CONEVAL) must define, identify, and measure poverty by taking into consideration at least nine indicators, one of which is "household access to basic services" [9]. There is no explicit indicator for energy poverty. However, as part of the indicators that measure multidimensional poverty, it considers lack of access to electricity and the type of cooking fuel. Both indicators can serve as possible variables to measure EP. García and Graizbord (2016) developed a method to measure energy poverty in Mexico in which a multidimensional index called "Pobreza energética en el hogar" is proposed [10]. This index showed that 11,093,000 households in Mexico (36.7% of households in the country), live in energy poverty. However, this method was not focused on the different climatic regions of Mexico; measurements were made at a state level, taking into consideration the climatic region of the whole state. On a positive note, it does consider thermal comfort and showed that this is one of the most deprived services in households, with 33% presenting a lack of it. In contrast, in analysis by Santillan et al., (2020) Mexico was analyzed by bioclimatic and climatic regions, demonstrating that Nussbaumer's index can adapt to the energy needs of each country without losing its essence [11]. We modified Nussbaumer's index by adding an additional dimension, thermal comfort, because this is both an important indicator mentioned in the literature [12] and also as an indicator with greater deprivation levels in Mexico, according to the study carried out by García and Graizbord [10]. We used data from the 2016 edition of the National Income and Expenditure Survey by INEGI [13]. The temperature of the region was considered to assess more closely to the real needs of people with respect to thermal comfort. In this way, we have two visions, one where temperature does not influence needs and one where it does, showing very interesting results, both in incidence and intensity in each region. It is expected that this regionalization will allow government entities to develop efficient energy policies. This will in turn foster a better understanding of energy poverty in Mexico to enable the planning of effective actions in short, medium, and long term so that everybody can have a better quality of life, with sustainable, efficient, and fair access to energy.

This paper will address energy deprivation using the multidimensional energy poverty index framework. In Section 2 we present the MEPI, climatic and bioclimatic regions, thermal comfort as a dimension, and temperature as the criteria to address the latter. In Section 3 we highlight the contrast of applying our modified proposal to both regionalizations. Finally, in Section 4 we address the importance of our results and the need to use a multidimensional energy deprivation index, and the need of further work in regard to more context related assessments on EP.

## 2. Materials and Methods

### 2.1. Multidimensional Energy Poverty Index

As mentioned earlier, Nussbaumer et al. (2012) developed the multidimensional energy poverty index (MEPI) that was applied in Africa. According to the citation analysis of the web of science, it has been cited 1109 times, being also applied and modified by Africa [7], Ecuador [14], the Philippines [15], and seven countries in Latin America [11], including Mexico (however, it was not used at a regional level) to measure its energy poverty. The MEPI is the product of a personal index (proportion of people identified as energy poor) and the average rate of those who fall under this category. Formally, the MEPI measures energy poverty in d variables in an n population. The MEPI considers a home as having energy deprivations if, for example, it does not have access to a kitchen that uses modern cooking fuels (LP gas, electricity, etc.) or does not benefit from the energy services provided by electricity. So, a person is identified as energy poor if the combination of the deprivations faced exceeds a predefined threshold [7]. The MEPI has been widely used and modified according to the energy needs of each country. One of the great advantages of the MEPI is the ease of decomposition, since the input data is at the micro-level (households or individuals), which allows expanding to a range of analyses focused on subgroups (socioeconomic, regional, etc.).

In the results generated when using the MEPI, the value closest to one indicates severe energy poverty, and the value closest to zero represents an absence of energy poverty. The examples we can use are Egypt, with an MEPI value = 0.01, compared to Mozambique and its MEPI value = 0.87, this means that Mozambique is much more energy poor than Egypt. Within this method, indicators are used and as the IEA (International Energy Agency) mentions, indicators are not simply data, they are essential tools that help link energy issues with sustainable development so that the public and politicians can promote an institutional dialogue [16]. For the selection of the indicators, Nussbaumer et al. (2012) used the Demography and Health Survey (DHS); in this study the variables were taken from the 2016 National Survey of Household Income and Expenditure (Encuesta Nacional de Ingreso y Gasto en los Hogares, ENIGH). Table 1 shows the variables and dimensions used to calculate the MEPI. The MEPI captures a set of energy deprivations that affect a person, and is made up of five dimensions that represent basic energy services with six indicators. The MEPI is actually made up of two things, an incidence measure (a proportion of people identified as energy poor) and an intensity quantification of energy poverty.

**Table 1.** Dimensions and variable cut-offs points defined by Nussbaumer et al., (2012) as well as the added thermal comfort dimension, variable and cut.off point (in italics).

| Dimension | Indicator | Variable | Deprivation Cut-off (Poor if … ) |
|---|---|---|---|
| Cooking | Modern cooking fuel | Type of cooking fuel | Use any fuel beside electricity, LPG, kerosene, natural gas, or biogas |
| | Indoor pollution | Food cooked on stove or open fire | True |
| Lighting | Electricity Access | Has access to electricity | False |
| Services provide by means of household appliances | Household appliance ownership | Has a fridge | False |
| Entertainment/education | Entertainment/education appliance ownership | Has a radio or television | False |
| Communication | Telecommunication means | Has a land line or a mobile phone | False |
| *Thermal comfort* | *Thermal comfort access* | *Has an air conditioner or heating* | *False* |

According to Fell (2017) and their analysis on 187 papers that included "energy service", there are 27 definitions of energy service and 173 examples. Some of those examples were found in 15 different sources, and are: lighting, cooking, heating, space heating, water heating, and refrigeration [17]. Also, Nussbaumer et al. refer exclusively to household needs, clearly stating that there are other energy needs required for societies to move forward. The most common energy services that are also included by Fell are lighting, cooking, entertainment/education, and services provided by means of household appliances and telecommunications. Nussbaumer et al. does not consider space heating/cooling due to the lack of available data. Cooking was chosen as one of the most basic energy needs and in relation to energy poverty the type of fuel was considered (firewood, charcoal, dung, etc.), since it can represent a health hazard. We know that energy access is crucial for community and personal development and that it is required in areas such as education, communication, and entertainment as well. As an energy use, electric household appliances like refrigerators are important in conservation of food and medicines.

This method offers the advantage of focusing on energy services and relying on data related to energy deprivation, rather than obtaining information through correlated variables (for example, energy consumption expenditure). Another advantage of the methodology that Nussbaumer et al. (2012) mentioned, is the ease to break it up, since the data used is at a micro-level, which means, households or individuals, and it can be modified or expanded by adding socioeconomic levels, regions, etc. Here is where the climatic regions are introduced, as mentioned earlier, Mexico is a very climate diverse country. Applying the method at a national level does not give the opportunity to fully understand the energy needs of people. However, if the method is applied at different levels, you will be able to see the needs of the people living in the north of the country and the needs of the people living in the south. Regionalizing will allow you to know the energy needs according to the different climates and characteristics of the region, adjusting the results to assess people's real energy needs.

### 2.2. Climatic Regions

Mexico is a country surrounded by two different oceans, with a geographical location between temperate and tropical areas of the planet, which makes it difficult to have a single climate or a single climate region. The term climatic region or province "refers to an extension of the earth's surface in which, due to its geographical location and landform orientation, the same wind systems dominate, and due to its latitude it has similar heating conditions, hence it shows great similarity in the types of climate, mainly in terms of rainfall regime, annual temperature movement, and thermal oscillation" [18]. Differences in altitude, exposure to winds, the amount of rain, and the value of the temperatures can cause the same region to have different degrees of humidity and temperature. This tells us that there is a wide diversity of climates and that each region has different energy needs and capacities [18]. The regions for this work were taken from the book "The Climatic Regions of Mexico", written by Rosalia Vidal Zepeda [18]. Eleven climatic regions have been identified: Northwest, Gulf of California, Central Pacific, North, Center, Northeast, Gulf of Mexico, Balsas River Basin and Valleys of Oaxaca, South Pacific, Southeast, and the Yucatan Peninsula. Figure 1 shows the climatic regions in Mexico.

### 2.3. Bioclimatic Regions

It was considered opportune to consider the bioclimatic regions due to the great variety of climates that Mexico has, since it is divided by the Tropic of Cancer and it clearly has two differentiated thermal zones. Bioclimates were considered to better understand the relationship with the dimensions of the MEPI, since bioclimatic regions are defined in close relationship with the weather characteristics of a locality, in comparison with climatic regions which pose a more general classification and consider first the political division (Mexican states) and their similar climatic characteristics. Given the importance of thermal comfort as a fundamental element when considering energy consumption, a

study of energy consumption in buildings in Mexico was consulted and it was confirmed that climate is more strongly linked to energy consumption than the level of economic development [18]. On the other hand, the importance of temperature is essential within bioclimates [11]. This indicates that measurement by climatic and bioclimatic regions is more accurate than by expenses and income, since energy needs are linked more to where you live.

**Figure 1.** Climatic regions in Mexico (Vidal Zepeda 2005 modified by Granados et al., 2007).

Due to the different elevations of the mountain ranges and the regions close to the coastlines, there are areas with extreme temperatures, whether they are desert climates or very humid climates. Bioclimate consists of determining the conditions or thermal sensations for human beings, like cold, heat, humidity, etc., in each ecological zone of the country [19]. For the general characteristics of bioclimates, Calixto and Huelsz [19] took information produced by the Instituto del Fondo Nacional de la Vivienda para los Trabajadores (INFONAVIT) and the Comisión Nacional del Fomento a la Vivienda (CONAFOVI), which were based on King (1994) and Morillón (2004 and 2005). Three types of bioclimate have been identified: semi-cold, temperate, and warm. These are further categorized by ambient humidity, dry, semi-humid, and humid, resulting in ten bioclimatic regions defined by Infonavit as: hot dry, extreme warm dry, semi-humid warm, warm humid, temperate, temperate humid, temperate dry, semi-cold, semi-cold dry, and semi-cold wet [20]. INFONAVIT published a list of the country's municipalities presenting each of the bioclimates [21], this facilitated the distribution of the data. The Figure 2 shows the bioclimatic regions in Mexico.

*2.4. Thermal Comfort*

Mexico is a country with lots of climate diversity. Although the climates are not as cold as they are in Europe, there are extreme climates and, due to climate change, they vary more and more. In Europe, thermal comfort is essential because there are cases of people dying due to the lack of it. Thomson et al. [2] estimated that in the summer of 2003 there were 70,000 deaths due to excessive heat.

Energy poverty, as mentioned before, occurs when a home is unable to access adequate levels of energy services, and thermal comfort is an energy service that must be considered due to the variety of existing climates and the radical change in temperatures in these last years. Space heating and cooling are among the problems of deprivation of energy services identified by Bouzarovski and Petrova [12]. Space heating is a problem for homes located in a cold climate, either because they are a low-income home or because they have

very limited energy services, and access to more efficient methods of domestic heating is a central issue for developing countries including Mexico [12]. This being said, the possibility of having access to space cooling in homes located in climates with hot summers, where currently heat waves have increased due to climate change, can be a problem [12]. The MEPI method is capable of accepting new dimensions and variables, Table 1 shows in bold the dimensions our methodology adds to complement the MEPI.

**Mexico: Distribution of climatic zones by municipalities**

**Figure 2.** Bioclimatic regions (CONNUE, 2014).

### 2.5. Temperature

It is necessary to consider a region's temperature range to understand whether or not a home needs appliances to achieve thermal comfort. Each region has meteorological stations in its respective municipalities that stores lots of temperature data. Only the stations that are in operation were considered (some municipalities have stations that are suspended). The stations closest to the municipality were used to calculate temperature. The temperature calculation was carried out at a municipality level, since not all of the towns in the municipality have a weather station. Annual minimum, maximum, and average temperatures were considered. It is important to mention that the data from the stations were provided directly by the Comisión Nacional del Agua (the National Water Board) [22]. According to Fuentes-Freixanet [23], the base temperature does not refer to comfort temperature, but to the temperature in which the air conditioning equipment begins to operate, in addition to the base temperature that varies from city to city. In this sense and for illustrative purposes, the base temperature was defined as 25 °C, heating temperature as 10 °C, and the ASHRAE standard for air conditioning or fan temperature was consulted, resulting in 30 °C. Simply put, heating is needed when temperatures are below 10 °C and air conditioning or fans are needed when temperatures are above 30 °C. Equation (1) (Heating Temperature) shows the distribution needs, where $C$ is heating, $t_{prom}$ is the average temperature, and $t_{min}$ is the minimum temperature.

$$C = \begin{cases} t_{prom} < 10\,°\text{C},\ C; \\ t_{min} < 10\,°\text{C},\ C \end{cases} \tag{1}$$

To know if air conditioning or a fan is necessary the same idea is used, where *AC* is air conditioning/fan, $t_{prom}$ is the average, and $t_{max}$ is the maximum temperature. Equation (2) (Air conditioning temperature).

$$AC = \begin{cases} t_{prom} > 30\,°C, \ AC; \\ t_{min} > 30\,°C, AC \end{cases} \tag{2}$$

## 3. Results

The MEPI was calculated for the eleven climatic and ten bioclimatic regions with data available from the ENIGH (National Survey on Household Incomes and Expenses in Spanish) and the multidimensional energy poverty limit was established as k = 0.3, defined by Nussbaumer et al. as implying that a person is considered energy poor if the sum of their deficiencies exceed the limit, whether they do not have access to a kitchen that uses modern fuels for cooking or do not benefit from the energy services provided by electricity. The regions are classified according to the degree of energy poverty, from acute energy poverty (bioclimatic MEPI > 0.06 for example, warm semi-humid), up to moderate energy poverty (bioclimatic MEPI < 0.04; for example, warm dry). Santillan et al. (2020) showed that a MEPI calculation with k = 0.3 as cutoff is representative for Mexico (as well as six other Latin American countries). The group tested different values for k. They found that for lower values of k, the number of persons living in energy poverty increases, but the intensity decreases and vice versa. Their conclusions on the pertinence of using k = 0.3 are sound and we applied their criteria to our study. Our main contribution lies in the inclusion of thermal comfort and the need to do regional assessments on this factor.

Figure 3 shows the intensity–incidence and MEPI relationship of the climatic regions of Mexico, using the dimensions and weights that Nussbaumer defines. It is observed that the regions that are closer to 1 suffer from greater energy poverty than those that are closer to 0. For example, the Gulf of Mexico suffers from a greater energy poverty than the Northwest, and not only that, its incidence rate (number of people living in EP) is higher. This tells us that in the Gulf of Mexico there are many poor people with too many deficiencies, unlike the Northwest, where there are only a few poor people, but those few have many deficiencies due to their intensity. Table 2 indicates the weights that were taken to calculate the MEPI.

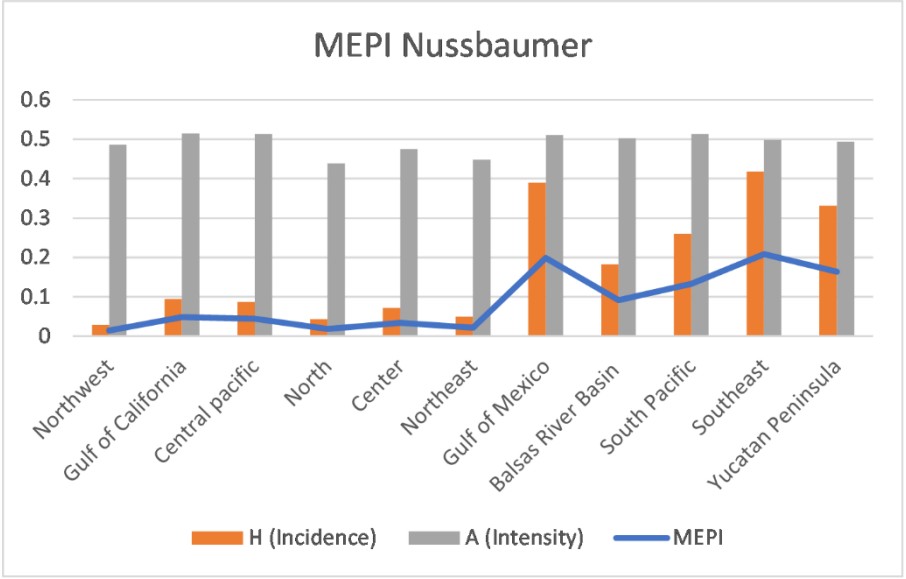

**Figure 3.** The MEPI in climatic regions of Mexico.

**Table 2.** Nussbaumer et al. weights.

| Dimension | Weight |
|---|---|
| Cooking | Modern cooking fuel 0.2 |
| | Indoor pollution 0.2 |
| Lighting | 0.2 |
| Household appliances | 0.13 |
| Entertainment | 0.13 |
| Communication | 0.13 |

Further MEPI calculations were performed by changing the weights of the variables and adding the thermal comfort dimension, the original Nussbaumer weights were also modified to include thermal comfort in such a way as to be proportional to the original. Table 3 shows the weights that were considered (a description and analysis of weights can be found in Appendix A). The weights were defined by a group of experts in energy (public policy, poverty, prospective, planning, and its social demand). The one-hour workshop started with the socialization of the methodology, followed by an individual classification and hierarchization of each dimension. With each expert's information, we applied normalization algorithms and demonstrated the results. In this case, the consensus of the group considered that cooking was the most important dimension, followed closely by electricity. The least important dimensions were communication and entertainment. It is important to comment that this analysis was done pre-COVID-19. As we have seen, the pandemic and subsequent contingency measures put in place have proved that having energy for communication and entertainment reasons in the household is critical.

Taking these weights into consideration, the MEPI calculation was performed for climatic regions considering thermal comfort and considering temperature and no temperature. In other words, the weights used by Nussbaumer will be used without any change (Table 2). The weights shown in Table 3 will be used considering thermal comfort and calculating the need of thermal comfort-related appliances in regard with the region's temperature (NTC, for Nussbaumer's weights with the addition of thermal comfort; TC, for weights defined by the panel of experts), in the following way:

If $t_{min}$ or $t_{prom} < 10\ °C$ then thermal comfort implies the need for heating in that household.

If $t_{max}$ or $t_{prom} > 30\ °C$ then thermal comfort implies the need for air conditioning.

**Table 3.** Thermal comfort weights.

| Dimension | Thermal Comfort Weight | Nussbaumer Comfort Weight |
|---|---|---|
| Cooking | Modern cooking fuel 0.13 Indoor pollution 0.13 | Modern cooking fuel 0.18 Indoor pollution 0.18 |
| Lighting | 0.24 | 0.18 |
| Household appliances | 0.21 | 0.115 |
| Entertainment | 0.07 | 0.115 |
| Communication | 0.08 | 0.115 |
| Thermal comfort | 0.14 | 0.115 |

In order to compare the different indexes, the same weights shown in Table 3 will be used, without considering temperature (°C). This means that the conditional explained above would not be used.

Figure 4 shows the difference in the thermal comfort MEPI with temperature (MEPI NTC) and without temperature (MEPI N). Between the MEPI with Nussbaumer weights with and without thermal comfort, a great difference is seen. For example, in the Gulf of California the MEPI without considering comfort is 0.04, very close to zero, which tells us that energy poverty is low, while when thermal comfort is considered, its MEPI becomes

0.09. Although the difference is minimal between tenths, it makes it clear that there are homes where the use of thermal comfort is necessary, and they do not have it. This is thanks to the established temperature standard, which tells us who needs air conditioning or heating but does not have it, or who has air conditioning but does not need it. In addition, if we look at Figure 4, in the "Incidence" graph the number of energy poor people increases considerably when thermal comfort is considered, meaning that there are more people who do not have air conditioning or heating when they need it. Another interesting case is the Northwest region which has an MEPI very close to zero, 0.014. When thermal comfort is considered, this becomes 0.067. This indicates that the number of energy poor people increases when considering thermal comfort; that is, when heating or air conditioning or both are required in homes. When observing the incidence of people who are energy poor, and the increase from 0.029 to 0.15, we can see that even though there are very few poor people, those few poor people have a great need of thermal comfort related appliances that is unfulfilled.

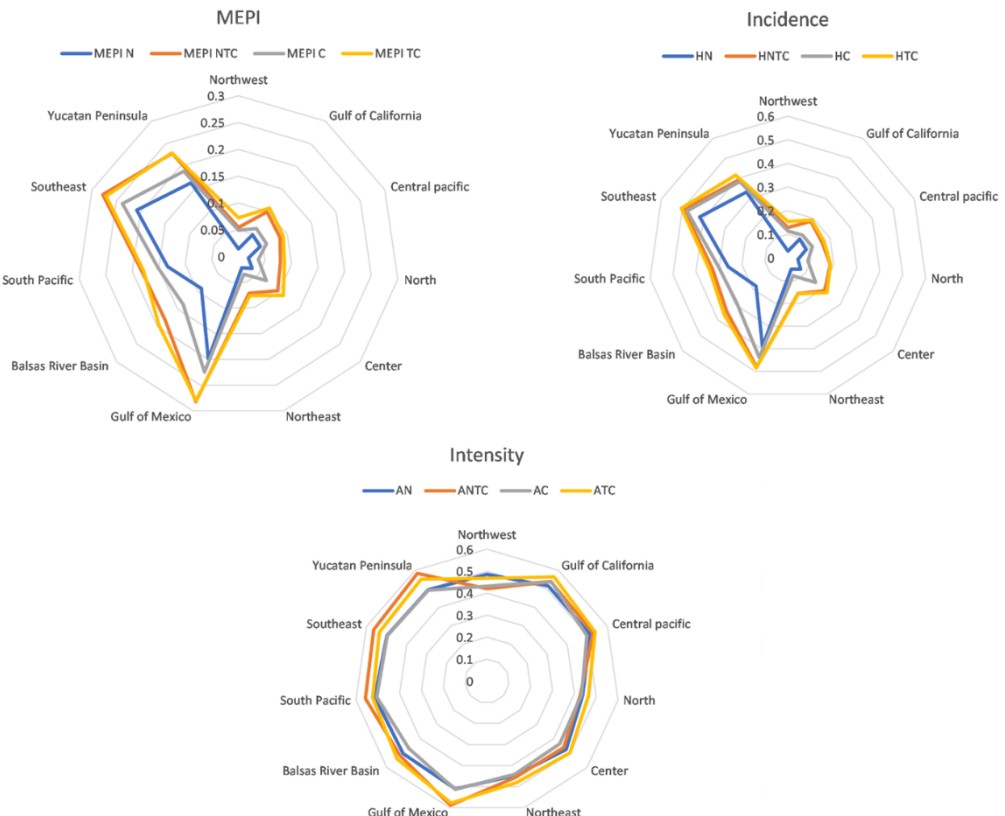

**Figure 4.** MEPI, Incidence (H), and Intensity (A) in climatic regions.

In Figure 4 the "Intensity" graph shows how poor the people are who are considered energy poor. Taking into consideration the previous examples, within the Northwest region the intensity decreases when considering the temperatures within the original Nussbaumer weights, from 0.48 to 0.42. This shows that by taking the temperature factor into account it is possible to identify where thermal comfort is really necessary and where it is not. In the case of the Central region, the intensity also decreases slightly, from 0.47 to 0.45.

When comparing the bioclimatic regions, it can be seen that the MEPI when using the original Nussbaumer weights and the modified Nussbaumer weights with thermal comfort the results are similar to the weights defined only for thermal comfort, only varying a little in extreme hot dry, semi-cold humid, temperate humid, and warm dry regions and varying more in semi-cold and temperate regions. In fact, the range changes, the bioclimatic MEPI goes from 0 to 0.25, and the climatic MEPI goes from 0 to 0.35. It can

be inferred that the bioclimatic MEPI has a more exact consistency due to its regions in comparison to the climatic MEPI, because the temperatures better match the definitions of the bioclimatic regions.

Figure 5 shows the comparison of the MEPI using the thermal comfort weights, the original Nussbaumer weights, and the MEPI with modified Nussbaumer weights. It can be seen that the MEPI with the original Nussbaumer weights increases when the thermal comfort dimension is added: a fair increase which clearly states that people are more energy poor when you take into account whether or not they have heating or air conditioning in addition to their needs. For example, the semi-humid warm region has an MEPI of 0.15, but when comfort is added considering the temperature the MEPI increases to 0.23. In all regions the MEPI increases when thermal comfort is added considering the temperatures.

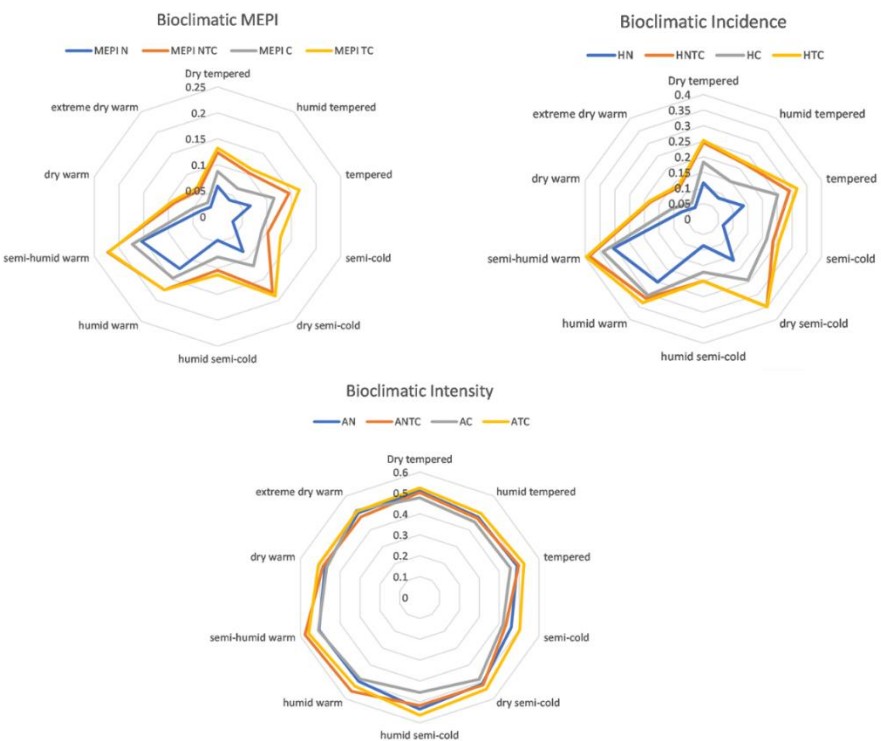

**Figure 5.** Bioclimatic MEPI, Incidence (H), and Intensity (A) with and without temperature considered.

In Figure 5 the "Bioclimatic Incidence" graph shows energy poverty considered in bioclimatic regions. A similar form is seen in Nussbaumer's incidences with thermal comfort considering the temperature (HNTC, for Nussbaumer's weights with the addition of thermal comfort; and HTC for weights defined by the panel of experts). Only in semi-cold and temperate regions is there a slight increase of the incidence considering comfort and temperature, while the incidence of the normal Nussbaumer is closer to zero than the incidence of the Nussbaumer with comfort. Extreme dry warm regions display similar results, from 0.045 to 0.063. However, the semi-cold region increases considerably from 0.06 to 0.21.

In Figure 5 the "Bioclimatic Intensity" graph shows how poor the people are who are considered energy poor. For the original weights defined by Nussbaumer and Nussbaumer modified weights considering thermal comfort with temperature (AN, for Nussbaumer's weights with the addition of thermal comfort; and ANTC for weights defined by the panel of experts), the intensity increased similar to the semi-humid warm region, from 0.5 to 0.58, and in the humid warm region from 0.49 to 0.55. Likewise, in the semi-cold region there is a slight decrease when thermal comfort and temperature are considered, from 0.46 to 0.44. Also, in cold semi-humid and extreme dry-warm regions, where there is greater incidence (in comparison with other regions), there is a better distribution of energy

services. Considering the weights in thermal comfort with and without temperature (AC, ATC), there is great similarity in the results of the extreme dry-warm region, from 0.514 to 0.508, while all the other regions showed an important increase when temperature was considered. This might show that energy poor people do not have access to air conditioning or heating when they need it, since this is the main reason for the increase in the intensity.

Figures 6 and 7 show a representation of the MEPI distribution with the original weights designed by Nussbaumer et al., as well as considering thermal comfort (including the temperature conditional on that behalf) in the different bioclimatic regions. Since the survey we used does not include all the municipalities, those regions are not shaded.

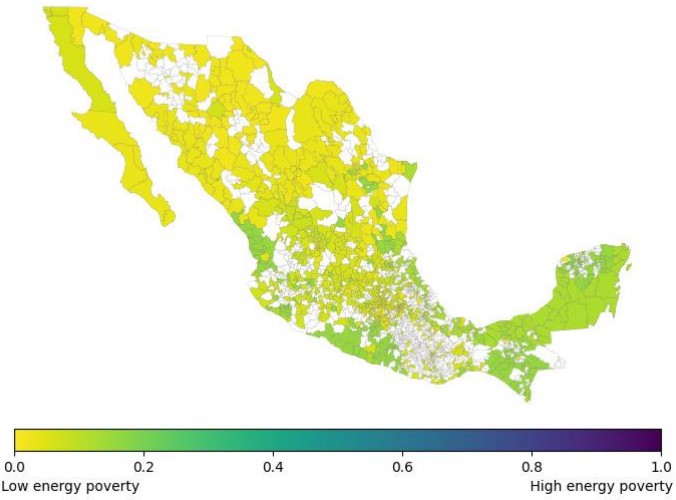

**Figure 6.** Bioclimatic MEPI.

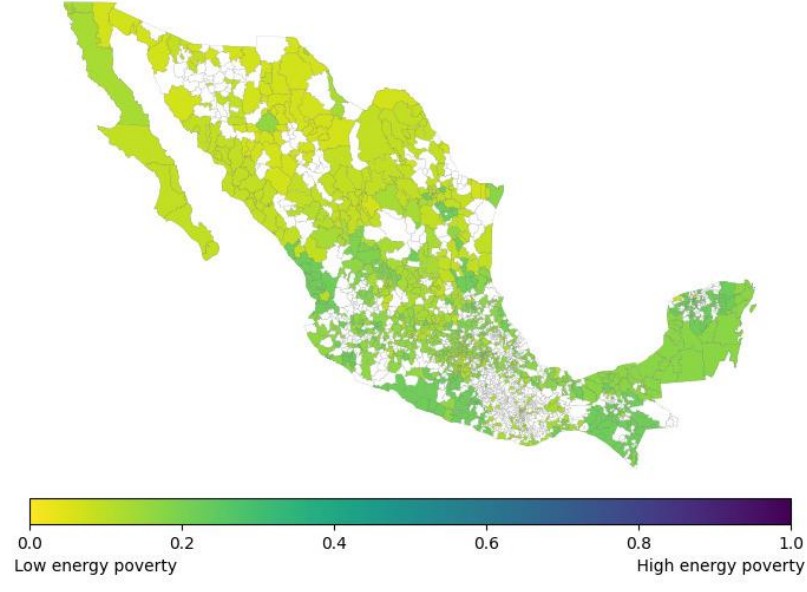

**Figure 7.** Thermal comfort (TC) bioclimatic MEPI.

## 4. Conclusions and Recommendations

The MEPI was applied to the different climatic and bioclimatic regions in Mexico and through the easy decomposition that MEPI provides one more dimension was added,

thermal comfort, and the weights of the variables were modified. When making use of the base temperature, heating temperature, and air conditioning temperature, it is observed that energy poverty becomes more generalizable than when these temperatures are not included, concluding that the thermal comfort factor has a significant impact. Not only that, it is possible to see that when considering thermal comfort when using the original Nussbaumer weights and those modified in all regions, the number of poor people increases, which means that more people suffer from this deprivation. However, in terms of intensity, the quantification of poverty lowers in regions such as the Northwest, the Balsas River Basin, Center, Central Pacific, and the South Pacific, which means that the number of people considered energy poor decreases when thermal comfort is considered; only people who really need comfort and do not have it are reflected here.

The results of this measurement can be considered as an instrument for government entities to create public policies that help mitigate energy poverty where people suffer the most. It is important to consider the region and climate rather than their income because energy takes natural factors into account. In addition, it is important that the current federal administration establishes its commitment to address energy poverty. This has been addressed in their most recent official documents, especially in the Estrategia de Transición para Promover el Uso de Tecnologías y Combustibles más Limpios (Transition Strategy to Promote the Use of Cleaner Technologies and Fuels), published in the Diario Oficial de la Federación (the official Mexican legal publication) in February 2020. We are currently providing valuable information to government officials on how to assess EP with different approaches. EP's complexity requires not only one, but several metrics and indicators to address it properly. We understand that our approach is centered on the deprivation of energy services, and this deprivation is highly dependent on context. The energy services related to thermal comfort are strongly dependent on geography, as addressed in this paper, but also to construction methods, cultural practices, and household dynamics. We believe that our work will establish an important step towards the construction of a multi-dimensional energy deprivation index (MEDI) that might better reflect the type of public policies that should be enforced to alleviate EP, and also, a straightforward tool for the evaluation of the impact of such policies.

As observed in the results, the MEPI is flexible and easily decomposed, since one more dimension was added, thermal comfort, and the methodology did not change. It could continue to be modified, adding new dimensions, new variables, new weights, or even updates to the dimensions that are already defined by Nussbaumer et al. (2012). For example, the education/entertainment dimension only takes into account if a household has a radio or a TV, but this could be expanded to add if they have a computer or an internet connection. This would also add to the education dimension, since radio and TV classify more as entertainment. Additionally, new dimensions totally different from those already defined could be created, obtaining more accurate results according to the needs of each region or country. We believe that every country and/or region should develop a context-related MEDI in order to achieve a better understanding of the particularities of EP in their populations. This would be helpful in the design of more adequate projects, programs, and public policies to address EP in a more comprehensive and sustainable way. The latter can be achieved by the prioritization of, in close collaboration with the affected population, the energy deprivation dimensions by impact, feasibility of alleviation, and perceived importance by the persons suffering from it. Thus, achieving more context related measures, outcomes, and impacts.

**Author Contributions:** Conceptualization, T.R.-B. and K.G.C.; methodology, K.G.C. and T.R.-B.; software, T.R.-B.; validation, T.R.-B. and K.G.C.; formal analysis, T.R.-B. and K.G.C.; investigation, T.R.-B.; resources, T.R.-B.; data curation, T.R.-B.; writing—original draft preparation, T.R.-B.; writing— review and editing, K.G.C.; visualization, T.R.-B.; supervision, K.G.C. All authors have read and agreed to the published version of the manuscript.

**Funding:** This research received no external funding.

**Institutional Review Board Statement:** Not applicable.

**Informed Consent Statement:** Not applicable.

**Data Availability Statement:** Publicly available datasets were analyzed in this study. This data can be found here: https://www.inegi.org.mx/programas/enigh/nc/2016/.

**Acknowledgments:** We want to express our appreciation to Manuel Martinez, Karla Ricalde, and Oscar S. Santillan for their help in the definition of the dimension's weights and their valuable comments. We also want to thank the National Science and Technology Council (CONACYT, in Spanish), for the scholarship awarded to the first author.

**Conflicts of Interest:** The authors declare no conflict of interest.

## Appendix A. Description and Analysis of Weights

One of the most challenging tasks when addressing any modification to the MEPI lies in the definition of the cut-off value, k, and on the weight for each dimension and variable. In the original proposal made by Nussbaumer et al. the sensitivity of k is addressed as well as the subjectivity on the selection of the weight. In this regard, for the definition of k, we used the threshold defined by Santillan (2020) and made an analysis on the variation of MEPI according to the value of k, as is shown in Figure A1.

Figure A1 shows that for values of k in the range of 0.2–0.3 the value for the MEPI showed a stable behavior resulting in a value around 0.8. As Nussbaumer et al. explain, a cut-off value of 0.3 implies that the household is considered energy poor if it has no access to either clean cooking or electricity powered energy services (or both).

To include an additional dimension first, we analyzed the MEPI and the relative importance of the variables from the dimensions of cooking and lighting against the rest of the dimensions (0.6 vs. 0.4). Then, we analyzed the relative importance of cooking vs. lighting (0.4 vs. 0.2), and we designed a set of values that could resemble as close as possible their relative importances while adding a new dimension and its variable (thermal comfort). The resulting set of values maintained the relative importance between cooking and lighting and modified slightly the relative importance between cooking and lighting against the rest of the dimensions (0.54 vs. 0.46, considering that the second set has an additional value). This set of weights was called MEPI with thermal comfort (MEPI CT).

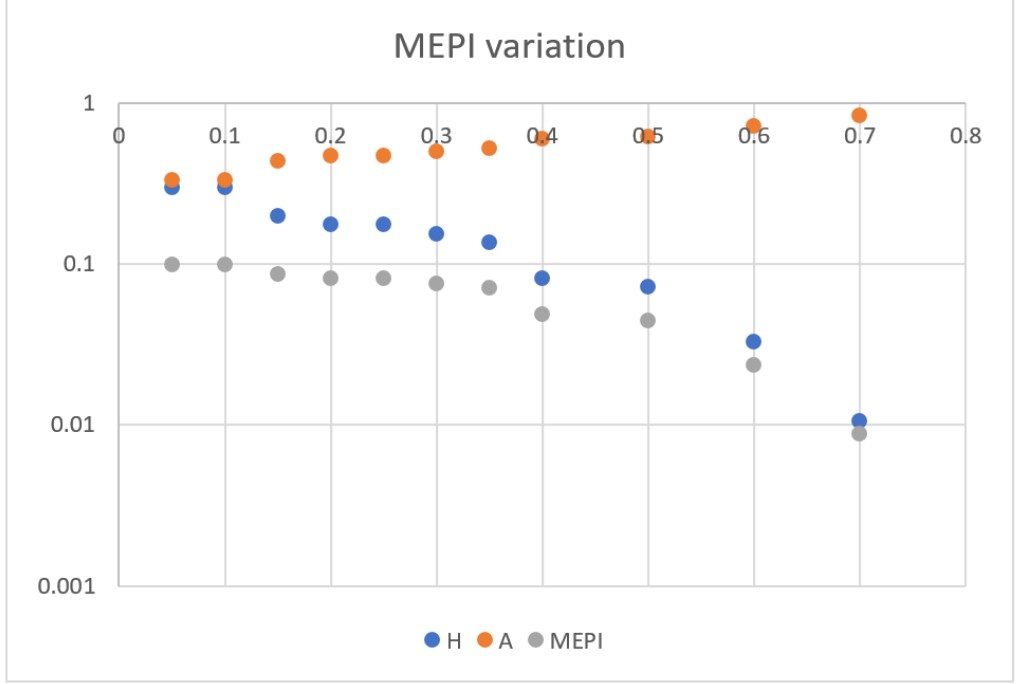

**Figure A1.** Variation of the MEPI for Mexico considering different values of k.

Finally, as we explained in Section 3, we used the same approach recommended by Nussbaumer et al. and organized a workshop with experts on energy use and demand in Mexico. All the experts ranked each variable on a scale from 1 to 9 (1, 3, 5, 7, 9). We obtained the average of each variable, and then normalized those averages. This led us to the final set of weights known as thermal comfort. This set does not maintain the relative importance between certain dimensions and variables that the MEPI has. As a matter of fact, the relative importance between cooking and lighting against the rest of the dimension is 0.5 vs. 0.5. We called this new set of values MEDI (multidimensional energy deprivation index) to avoid confusion with MEPI and MEPI CT.

Once we had set the weights, we needed to address the robustness of the cut-off value k for this new set of weights. We calculated the MEPI, MEPI CT, and MEDI for the ten bioclimatic regions (and the whole country) with varying values of k: 0.2, 0.25, 0.3, and 0.35 (the latter was a control value, since we had already seen the difference that 0.35 represented against the other three more consistent values). In Figures A2–A4, we show the results of such calculations.

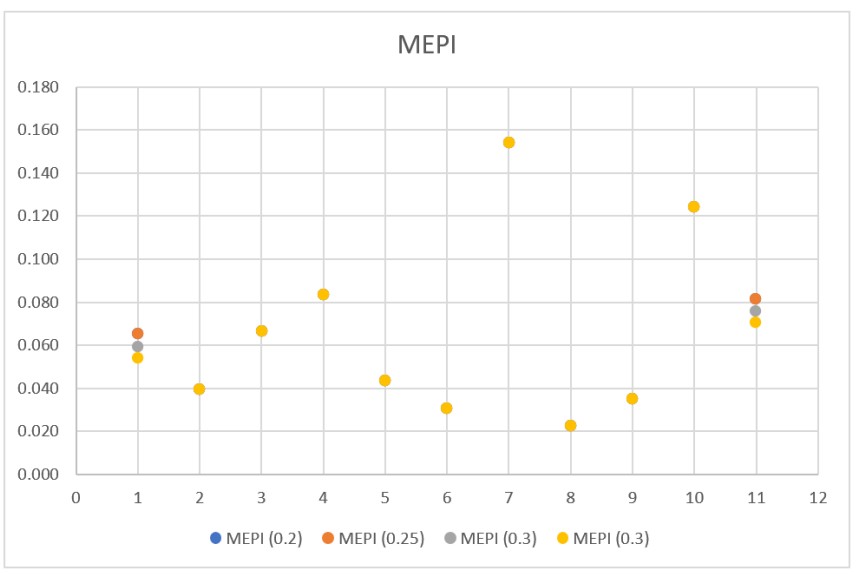

**Figure A2.** MEPI variations in the ten bioclimatic regions and Mexico (k = 0.2, 0.25, 0.3, and 0.35).

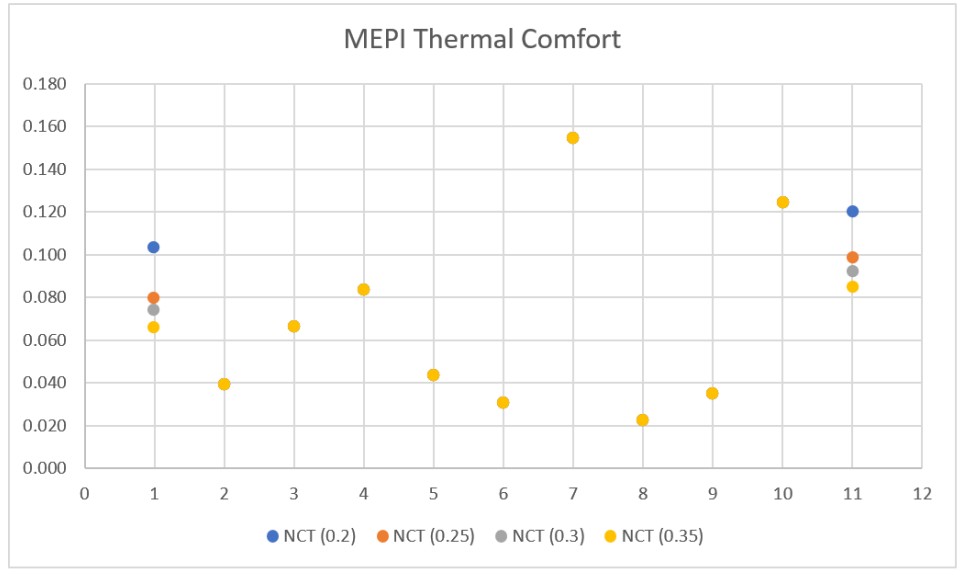

**Figure A3.** MEPI CT variations in the ten bioclimatic regions and Mexico (k = 0.2, 0.25, 0.3, and 0.35).

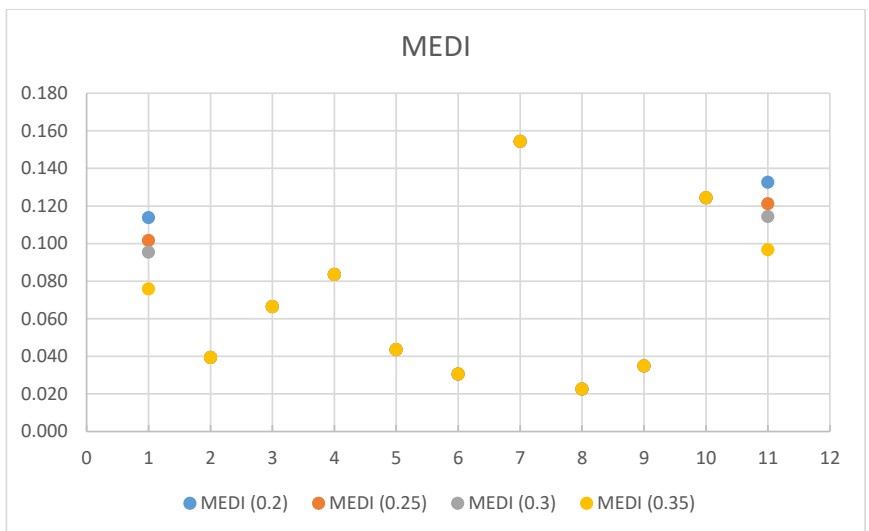

**Figure A4.** MEDI variations in the ten bioclimatic regions and Mexico (k = 0.2, 0.25, 0.3, and 0.35).

We can observe that for nine of the ten bioclimatic regions the cut-off value variation had no effect. However, for the region tagged with 1, and the value in the whole country (tagged 11), the least variation happens with k = 0.25 and k = 0.3. This was also shown by the very high correlation between the values of the Spearman and Kendall coefficients in Table A1, where we can see the stability of the bioclimatic region values that implies that the change in the cut-off marginally affects the MEDI value.

**Table A1.** Correlation in the bioclimatic regions MEDI when the value of k is changed [24].

| Spearman | 0.20 | 0.25 | 0.30 | 0.35 |
|---|---|---|---|---|
| 0.20 | 1 | | | |
| 0.25 | 0.9909091 | 1 | | |
| 0.30 | 0.9909091 | 1.0000000 | 1 | |
| 0.35 | 0.9818000 | 0.9909091 | 0.9909091 | 1 |

| Kendall | 0.20 | 0.25 | 0.30 | 0.35 |
|---|---|---|---|---|
| 0.20 | 1 | | | |
| 0.25 | 0.9636360 | 1 | | |
| 0.30 | 0.9636360 | 1.0000000 | 1 | |
| 0.35 | 0.9272720 | 0.9636360 | 0.963636 | 1 |

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
