# Peer review of "Addressing Thermal Comfort in Regional Energy Poverty Assessment with Nussbaumer’s MEPI"

_sustainability, doi:10.3390/su13010352_

Round 1

Reviewer 1 Report

Added in the PDF

Reviewer 2 Report

This is an interesting paper in which the authors capture the spatial variations in climate for energy poverty across Mexico – considering both cooling and heating services. Thank you for the opportunity to read your interesting work!

Your title

  • I think it is important to include “the capabilities approach” in your title, to make it easier for interested scholars to find your research.
  • It is also not clear to me what is meant by “climatic regions”. In your article you compare results across climatic regions in Mexico, and your title could reflect this better.

Your abstract

  • Although well written, a lot of your abstract is wider context and takes a while to get to what you will be doing in the paper, and its novelty.

Introduction

  • The introduction is well written and researched, giving a nice overview of different attempts to measure energy poverty. However, there is a lot of material in this section! It would be good to see a clear set of aims and a structure for the paper at the end of the introduction.
  • It would also be beneficial to move some of the detailed evaluation of Nussbaumer’s index into Section 2 (e.g. line 57-70).

Materials and methods

  • It would be good to see some more detail in this section to justify your approach. For example, its interesting to focus on “basic energy services” but perhaps you could provide further definitions, and discuss further the strengths and weaknesses, of this approach e.g. Fell, M. J. (2017). Energy services: A conceptual review. Energy research & social science, 27, 129-140.
  • In Section 2.2. as a geographer I would love to see a map to illustrate the climatic regions and bio-climatic regions.
  • Table 1 and Table 2 could be combined, but just make it clear (e.g. using shading) that thermal comfort is an extension to Nussbaumers index.

Results

  • It is fascinating to see your analysis account for both heating and cooling energy services, and the implications across different climates with their own specific challenges.
  • You use k=0.3 as defined by Nussbaumer, but for me this could benefit from further justification, is I appropriate to transfer k=0.3 between national contexts?
  • Your Radar Charts could be combined in one or two figures. It you also be lovely to see a map or two of the distribution of the results if possible!!

Conclusions and recommendations

  • The final sections on conclusions on recommendations could be combined, and strengthened. I would like to know what are the key implications of your results for energy poverty and policy in Mexico? And what are the implications of your results for other national contexts (e.g. similar/different)?

Minor points

  • Line 112: Please could you translate “Encuesta Nacional de Ingreso y Gasto en los Hogares” for those who can’t speak Spanish.
  • At risk of sounding really nit-picky (sorry!) for clarity I would try to avoid the over use of semi-colons (;)
  • “Section 6 “ Conclusion” should be Section 5.

Round 2

Reviewer 1 Report

Authors have done a very interesting work on energy poverty. It is very well structured and written.

However, I do not like the weights used. They mention that they have used the original Nussbaumer weights for their first measurement. Then, they have added a new variable, the thermal comfort, and they have modified the weights in a proportional way.

Firstly, the weights are not proportional. I understand that the main part of this work is the measurement of energy poverty with the Nussbaumer method and a new measurement by adding a new variable. For this, the main work of the authors is the choice of weights and the cut-off. For this purpose, the new weights and the new cut-off should be chosen carefully.

In fact, I wrote in my first review, I do not agree with the chosen weights and cut-off. In fact, they are comparing two very different scenarios. And It has no sense to compare the results for those weights. I have done an effort to revise this paper and to try to explain the problem with the weights. Nevertheless, authors have not changed them.

In my opinion this paper is very interesting and it is very well written but the weights are not chosen in a right way.

I told in my first review that they should change the weights. If authors do not change them, for me this paper will be a reject.

Author Response

We gratefully thank the Reviewer for the comment on weights and cut-off definition. We think that manuscript quality has improved after corrections. It was revised thoroughly, and corrections were carried out. We greatly appreciate the opportunity to amend the manuscript and hope to submit a better paper with the Reviewer’s suggestion which will be able to fulfil the standards and requirements of Sustainability. Our revised manuscript is accompanied with this letter were the main change introduced is listed below. Reviewer’s comments are in red. All corrections added in this second round to the manuscript were set in green.

Authors have done a very interesting work on energy poverty. It is very well structured and written.

However, I do not like the weights used. They mention that they have used the original Nussbaumer weights for their first measurement. Then, they have added a new variable, the thermal comfort, and they have modified the weights in a proportional way.

Firstly, the weights are not proportional. I understand that the main part of this work is the measurement of energy poverty with the Nussbaumer method and a new measurement by adding a new variable. For this, the main work of the authors is the choice of weights and the cut-off. For this purpose, the new weights and the new cut-off should be chosen carefully.

In fact, I wrote in my first review, I do not agree with the chosen weights and cut-off. In fact, they are comparing two very different scenarios. And It has no sense to compare the results for those weights. I have done an effort to revise this paper and to try to explain the problem with the weights. Nevertheless, authors have not changed them.

In my opinion this paper is very interesting and it is very well written but the weights are not chosen in a right way.

I told in my first review that they should change the weights. If authors do not change them, for me this paper will be a reject.

We are especially thankful for the reviewer’s noticing a mistake in table 3 (lines 276-277), we wrongly copied the weight for lighting (it should have been 0.18, not 0.115). The new weights--as well as the cut-off (the latter is not a new figure, it is the same defined by Nussbaumer and by Santillan and we show the strength of that selection for our particular case)--are explained and analyzed in the Appendix “Description and Analysis of Weights” starting in line 411.

The weights used for the case Nussbaumer with Thermal Comfort (NCT), include a proportional variation needed to include the new dimension, thermal comfort, and its variable. The calculations were done with the proportional weights, our mistake was a typing error, now it can be seen that the sum of the weights add to 1.0 (as it should be).

We are confident that the Appendix has the proper explanation on weights and cut-off without distracting the readers from the proposal of adding Thermal Comfort as a dimension, finding the correct variables to address that particular deprivation and, most importantly, developing an additional conditional regarding temperature to properly consider (or not) the energy need in the specific bioclimatic region.

Round 3

Reviewer 1 Report

I will accept in present form